# The Impact of Low Back Pain on the Quality of Life of Children between 6 and 12 Years of Age

**DOI:** 10.3390/healthcare11070948

**Published:** 2023-03-24

**Authors:** Elisiane de Souza Santos, João Marcos Bernardes, Luana Schneider Vianna, Carlos Ruiz-Frutos, Juan Gómez-Salgado, Melissa Spröesser Alonso, Matias Noll, Adriano Dias

**Affiliations:** 1Public/Collective Health Graduate Program, Botucatu Medical School, São Paulo State University (UNESP), Botucatu 18618687, Braziljoao.m.bernardes@unesp.br (J.M.B.);; 2Department of Public Health, Botucatu Medical School, São Paulo State University (UNESP), Botucatu 18618687, Brazil; 3Medical School, Centro Universitário de Jaguariúna, (UNIFAJ), Jaguariúna 13820000, São Paulo, Brazil; 4Department of Sociology, Social Work and Public Health, Faculty of Labour Sciences, University of Huelva, 21007 Huelva, Spain; 5Safety and Health Postgraduate Programme, Universidad Espíritu Santo, Guayaquil 092301, Ecuador; 6Health and Physical Examination Laboratory, Instituto Federal Goiano, Ceres 76300000, Goiás, Brazil; 7Physical Education Graduate Program, Universidade Federal de Goiás, Goiânia 74690900, Goiás, Brazil

**Keywords:** low back pain, quality of life, child, adolescent, cross-sectional studies

## Abstract

This study aimed to evaluate the impact of low back pain on the perceived health-related quality of life of children between 6 and 12 years of age. This is a cross-sectional study of three hundred seventy-seven students from three schools (two private and one public) located in the city of Botucatu, São Paulo. Data were collected using the Pediatric Quality of Life Inventory (PedsQL, version 4.0), a questionnaire comprising questions on personal background, sociodemographic and socioeconomic information, and a questionnaire about quality of life. Comparisons were made between groups with and without low back pain. The chi-squared test was used for analyzing categorical variables, and the non-parametric Mann–Whitney test was used for continuous variables. According to the findings obtained in this study, it was observed that low back pain in the last month was reported by 27.3% of the total participants. The perceived health-related quality of life was lower among individuals who had low back pain, and the scores of physical and emotional functioning domains were also lower in the presence of low back pain. The prevalence of low back pain among children and adolescents is relatively high. Furthermore, the repercussions of low back pain may lead to a lower overall perception of the health-related quality of life in this population and affect aspects of physical and emotional functioning.

## 1. Introduction

Low back pain includes pain, muscle tension, or stiffness located between the 12th rib and gluteal folds, and it may or may not radiate to one or both legs [1,2]. It can affect up to 90% of the adult population [3,4]. In children, the prevalence of low back pain can vary from 30% to 70% [5,6], depending on the age of the population in question, and it is higher among older children. According to a recent study on low back pain and risk factors in this population, low back pain has a higher prevalence (10.34%) compared to other regions of the spine, such as pain in the thoracic spine (5.75%) and cervical (2.3%) [7]. This scenario diverges considerably from the last decade’s literature as low back pain among children went from being a rare occurrence and a possible indication of a serious underlying disease to a quite common one with a benign clinical course [8,9,10,11].

The etiology of children’s back pain differs significantly from that of the adult population; most cases of low back pain in children are non-specific (i.e., without an evident underlying pathology) and self-limiting [12,13].

Regarding potential risk factors, it is common knowledge that low back pain is a multifactorial condition and numerous exposures, including nonmodifiable ones, have already been associated with a higher probability of its development among children. Overall, the findings from previous studies indicate that sociodemographic characteristics (sex, age, body mass index, not having siblings, not living with both parents, family history of back pain, family educational attainment, and family income), lifestyle habits (physical inactivity and backpack use), biomechanical factors (abnormal spinal curvatures, trunk muscle endurance, and linear growth), physiological changes (pubertal development), and psychosocial factors (adverse psychosocial exposure, peer relationship problems, and conduct problems) are associated with the occurrence of low back pain in children [6,14,15,16,17,18,19,20,21,22]. Nevertheless, it is important to highlight that the evidence regarding risk factors for low back pain in this population is sometimes inconsistent and conflicting, mainly due to the heterogeneity of the investigation proposals, designs, instruments, and differences in the exposures and/or outcome definitions and measurements [10,23].

Even if the children’s risk factors for back pain remain somehow unclear, its life-course trajectory is already being studied. It has already been shown that low back pain experienced in childhood may carry on into adolescence [9] and even into adulthood [24], when it becomes the seventh and fourth main cause of disability-adjusted life-years, respectively [25]. Thus, low back pain is acknowledged as a major public health problem among adults since it leads to social and personal losses, such as absences from work and limitations in the activities of daily living (ADLs) [10,26,27,28].

Among children, however, the importance of low back pain has been largely overlooked for decades [24]. It was only recently, when it became clear that low back pain is not a series of unrelated self-limiting events and that the occurrence of low back pain among children and adolescents increased the chances of presenting low back pain in adulthood, that this situation started to change [29]. Therefore, it is still not evident if the severe consequences of low back pain in adults are also common during childhood since the specific impact of low back pain among children has not been comprehensively studied yet. Even though some studies have found that children with low back pain experience a quality-of-life impairment, increased healthcare utilization, increased likelihood of regular analgesic use, reduced physical activity, and school absenteeism [6,11,19,30,31,32,33,34], more studies that investigate the consequences of low back pain in this population are in need, since the extrapolation of knowledge from adult low back pain research findings to children may not be advisable due to the significant physical, physiological, psychosocial, and lifestyle differences between childhood and adulthood.

Although the literature notes that psychosocial and emotional factors play important roles in the occurrence of low back pain [10] and can reduce the quality of life (QoL), few studies have investigated how low back pain impairs the perceived QoL among school-going children [35,36,37]. Because of the association between psychosocial factors and low back pain, and considering that these factors determine health-related quality of life (HRQoL), this study aimed to investigate the impact of low back pain on the perceived HRQoL in children and adolescents who are attending school.

## 2. Materials and Methods

### 2.1. Participants

This cross-sectional study integrated an umbrella project [31] in which 377 schoolchildren 6–12 years of age participated (Figure 1). The participants were recruited from three schools (two private and one public) located in the city of Botucatu, São Paulo State, Brazil. These schools were selected by simple random sampling, and the data were collected between February and December 2017. 

Located in the midwestern region of the state of São Paulo, Botucatu is a medium-sized city with approximately 140,000 inhabitants, including around 18,000 children in the age range of the investigation. This number determined an estimated minimum sample size of 317 participants, considering a prevalence rate of 30% to 70%, as well as type I and II errors of 5% and 20%, respectively [38]. Further information about how the sample size was calculated can be found in the previous study [31]. The study was developed after approval from the Research Ethics Committee of the School of Medicine of Botucatu (approval number: 1.782.512).

### 2.2. Procedure

After those responsible for the schools signed the consent form, meetings were held with the participants’ parents and/or guardians to present the research objectives and procedures, and to deliver the informed consent forms and the questionnaires on the participants’ personal and socioeconomic backgrounds. Participants with authorization from their parents and/or guardians to participate in the study were then invited to do so after being informed of the objectives and procedures. Finally, after signing the assent forms, participants answered the Pediatric Quality of Life Inventory (PedsQL, version 4.0) questionnaire and underwent anthropometric and kinesiologic measurements.

### 2.3. PedsQL

A previous study adapted and validated the PedsQL for the Brazilian population; Cronbach’s alpha value for the whole scale has been reported to be 0.88, while the individual domain values were between 0.6 and 0.85 (for school and physical functioning domains, respectively), demonstrating adequate internal consistency [39]. Although it has been in use for years, knowledge about other psychometric properties of the Brazilian PedsQL 4.0 version and appropriate validation for other pediatric populations in this country are still lacking.

The PedsQL contains 23 items divided into 4 domains—physical functioning (8 items), emotional functioning (5 items), social functioning (5 items), and school functioning (5 items). These domains are related to how much each item involved some level of difficulty in the last month based on 5 responses, and are scored as 0 (never), 1 (almost never), 2 (sometimes), 3 (often), and 4 (always). These responses are then converted to a scale of 0 to 100 (0 = 100, 1 = 75, 2 = 50, 3 = 25, and 4 = 0), and the average of the items corresponds to a measurement of the QoL. A higher score corresponds to a better QoL.

All participants were age eligible to self-report (ages between 5 and 12 years) and answered the PedsQL age-appropriate version (5–7 and 8–12 years) according to their age range to facilitate understanding of the instrument by participants from each age group. Although it is originally self-applicable, an interviewer applied the PedsQL to prevent data incompleteness [39,40].

### 2.4. Anthropometric and Kinesiologic Measurements

A trained physical therapist conducted the anthropometric and kinesiologic evaluations, which comprised the following variables: support base length, lumbar angle, waist-to-hip ratio, lumbar spine mobility, body weight, and height. A detailed description of how each of these variables was measured is available in another publication [31].

### 2.5. Variables

The variables analyzed in this study are presented below.

Continuous variables: age (measured in years); height (measured in cm); Body Mass Index (measured in kg/m^2^).

Categorical variables: sex (male and female); education (public and private); income (not sure, up to three minimum wages, and over three minimum wages); race (Caucasian and other); and physical activity (yes and no).

Scores for the PedsQL and their respective dimensions—physical, emotional, social, and school functioning (never, almost never, sometimes, often, always)—were used to measure the HRQoL.

To identify the prevalence of the outcome, the participants were classified as having low back pain if they indicated feeling pain in the lumbar region (below the ribs and down to the gluteal folds) on a body map adapted from the Nordic Musculoskeletal Questionnaire [41] during the last month [1,42]. It is important to note that only the adapted body map from the Nordic Musculoskeletal Questionnaire was used in this study in order to pinpoint the exact pain location.

### 2.6. Statistical Analysis

The means and standard deviations were calculated for the following variables: continuous and categorical. Next, the groups of participants with and without low back pain were compared. The chi-squared test was used for categorical variables, and the non-parametric Mann–Whitney was used for continuous variables, given the non-normal distribution of the responses. A *p*-value of < 0.05 was considered statistically significant. The Cohen’s d effect size, a standardized difference of means, was estimated to evaluate the magnitude of the difference in the HRQoL scores; the outcome (HRQoL) was significantly different when groups (with and without low back pain) were compared. The IBM/SPSS^®^ Statistics software package, version 25.0, was used to process the database and conduct the analyses.

## 3. Results

Of the 377 individuals, 103 reported experiencing low back pain, resulting in a prevalence of 27.32% (95% CI, 23.07–32.03).

Table 1 shows the participants’ sociodemographic characteristics. Concerning the anthropometric variables, the groups with and without pain were homogeneous among themselves, in addition to the sex variable. Females were the dominant sex in both groups (with pain (52.4%) and without pain (50.4%)); however, this was not statistically significant (*p* = 0.721).

As for the HRQoL, it is possible to observe lower scores in almost all of the domains of the PedsQL in the group with pain (Table 2). The values were statistically different between the groups for the domain related to physical functioning, in which the group with pain and the group without pain presented means of 80.1 (±15.7) and 84 (±13.9), respectively (*p* = 0.017).

For the emotional domain, mean values of 56.9 (±23) for the group with pain and 66.7 (±20.8) for the group without pain (*p* < 0.001) were found. In addition, a statistical difference was noted in the overall score with mean values of 73.6 (±14.9) for the group with pain and 78.5 (±12.7) for the group without pain (*p* = 0.008).

## 4. Discussion

According to the findings, the prevalence of low back pain in school-going children was 27.3%; despite being quite high from an epidemiological point of view, it is lower than that reported in a previous study [10]. This difference can be because the population in that study comprised children in higher age ranges and assumed a different definition to classify low back pain. Due to recent investigations of low back pain in children, factors such as prevalence [31,43,44], impairment in adult life [24,43,45,46], and annual medical costs [47,48] have become more known and recognized, but still need further investigation. Thus, with established definitions and a prevalence reported worldwide [4,10], it is extremely important to understand its impacts on the general health of this population so that more effective measures regarding prevention and treatment can be created.

Participants that reported experiencing low back pain presented a significantly lower HRQoL score. Previous studies have found similar results [8,18,49]; however, all of them investigated older participants and two of them [18,49] used different instruments to evaluate the HRQoL. The HRQoL is a subjective concept that can be influenced by specific factors, as well as by conditions that affect one’s health; thus, it is plausible that low back pain may play an important role in one’s perceived quality of life [8,40].

According to Cohen’s d effect size estimate, the current result shows a difference in the HRQoL of moderate magnitude [50] between children with and without low back pain. The obtained effect size of 0.34 may be expressed as a 34% probability that a randomly sampled child with low back pain will have a lower HRQoL score than a randomly sampled child without low back pain [51]. Thus, knowledge of one’s HRQoL may help with identifying children and adolescents with low back pain, and it may also be crucial to establish this population’s healthcare needs in a more thorough way. Even though previous research findings showed only a low effect of low back pain in the HRQoL [52,53], it is important to note that these studies’ participants were older and the HRQoL was evaluated with other instruments.

Likewise, the results have shown a significantly lower mean score in the HRQoL domain related to physical functioning in the group with low back pain, corroborating previous studies [49]. This difference between groups was also of moderate magnitude [50]. Since this domain contains questions related to the ADLs and there is evidence that functional status impairments in children with low back pain may be associated with decreases in the QoL [52], this finding may indicate that the lower score in the HRQoL physical functioning domain could be linked with the ADLs’ limitations in young individuals with low back pain [33,54].

If the above hypothesis is true, these possible ADLs’ limitations could be due to pain-induced behaviors. It is known that some individuals are afraid to either feel the pain or that it will intensify, and therefore they start to avoid certain activities [54]. This limitation signals a well-known phenomenon known as kinesiophobia. Another potential explanation for a possible limitation in the ADLs is that individuals who feel musculoskeletal pain adopt postures that lead to muscle tension and consequently decrease their range of motion, resulting in pain in other parts of the body and restrictions in certain ADLs [19].

The HRQoL emotional functioning domain has presented a significantly lower mean score among children with low back pain too. In addition, as with the total HRQoL and the physical functioning domain, this difference was of a moderate magnitude [50]. This domain included feelings, such as anger, depression, and even sleeping difficulties. Interestingly, all these factors have not only already been shown to be associated with low back pain [10,55,56], but the literature from the last four decades based on paradigms such as the biopsychosocial model shows that psychosocial factors, more so than mechanical factors, have a strong association with low back pain [10,43].

Finally, the HRQoL domains related to school and social functioning did not present statistically significant differences between the groups. The lack of association between low back pain and these domains could be explained by the fact that child and parental agreement on the occurrence of low back pain has already been reported to be of only 33% [57]; thus, children might not be allowed to be absent from school and/or social activities due to their complaints of low back pain.

Investigating the effects of low back pain in specific domains of self-perceived HRQoL may lead to intervention strategies best suited to children’s needs and more efficient health promotion programs among this population.

Although a blinded evaluator to investigate the outcome was not employed, which may be viewed as a limiting factor of the investigation, all the data collection, evaluations, and physical exams were standardized to minimize biases, similar to a recent study [58]. Another relevant point was the decision to obtain the outcome of low back pain through self-reporting. This occurred mainly because the study addressed low back pain in children, thus making the exclusive use of imaging tests infeasible. Moreover, it was also sought to prevent the nocebo effect, which tends to worsen participants’ health status when they receive a diagnosis through imaging exams [59,60,61] and leads to their increased use of medication, thus “catastrophizing” the pain [59]. Finally, even though the number of participants was almost 20% higher than the calculated minimum sample size, it is important to note that roughly only one-third of the eligible participants took part in the study. It is known that a low response rate might lead to selection bias, due to participants disproportionately possessing certain characteristics that might affect the outcome when compared to non-participants.

The strengths of the study include the random selection not only of the participants but also of the schools, which is expected to reduce the risk of the selection bias occurrence discussed above. However, it should be highlighted that it is known that the “play of chance” might result in unbalanced groups even with randomization, especially in the presence of a considerable exposure prevalence combined with a modest number of participants. Since it was not possible to compare the characteristics of the study’s participants with the source population (selection bias analysis) and non-participants (selective non-response bias analysis), to evaluate if there was a systematic difference in characteristics between them, it is not possible to completely rule out the existence of selection bias even with a random selection of participants. Another strength was the adoption of the PedsQL questionnaire, which has been translated, validated, and adapted for the Brazilian population [8], unlike most studies assessing the QoL in children. Lastly, the observed individuals were 6–12 years of age—this is a younger age range than those usually investigated in studies regarding the association between low back pain and the HRQoL among children.

Finally, given the fast-paced technological developments and swiftly changing lifestyles (i.e., the growing use of tablets and smartphones by adolescents and children in the last decade) that seem to be endemic to modern societies, a series of cross-sectional studies regarding low back pain and the HRQoL among children is a suggestion of future research, since it would offer insights into possible exposures and outcome changes in the future. Another interesting avenue of research is to examine how pain in other spinal sites and in multiple spinal sites affects the children’s HRQoL, since the prevalence of neck pain has been increasing in recent years, even though low back pain is still the most common type of spinal pain among children and adolescents [62].

## 5. Conclusions

Based on the results obtained in this study, low back pain has a relatively high prevalence among children and adolescents. In addition to its quite high prevalence, children with low back pain present a lower perception of HRQoL, especially in aspects related to their physical and emotional functioning, according to how the PedsQL measures those domains.

## Figures and Tables

**Figure 1 healthcare-11-00948-f001:**
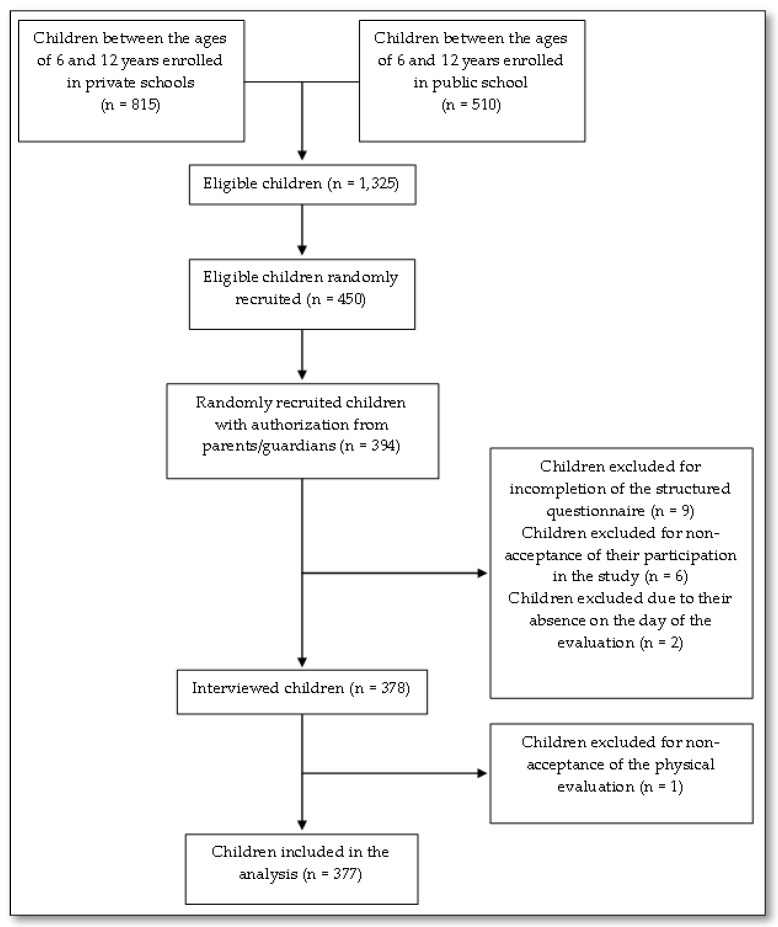
Flowchart of the Research Participants.

**Table 1 healthcare-11-00948-t001:** Sociodemographic profile of the research population, separated into two groups: with pain and without pain.

	With Pain		Without Pain			*p*-Value ****
	Mean	SD *	Mean	SD *	Mean Diff.	
Height	1.39	0.12	1.37	0.13	−0.02	0.223
Age	8.85	1.83	8	1.76	−0.85	0.006
Income (in minimum wages)	*n*	%	*N*	%		
Up to 3 MWs	19	47.5	58	54.2	--	0.225
Over 3 MWs	20	50.0	46	43.0	--	
Not sure	1	2.5	3	2.8	--	
Education	*n*	%	*N*	%		
Private	56	54.4	111	40.4	--	0.015
Public	47	45.6	164	59.6	--	
Race	*n*	%	*N*	%		
Caucasian	93	90.3	256	93.4	--	0.679
Other	10	9.7	18	6.6	--	
Sex	*n*	%	*N*	%		
Female	54	52.4	138	50.4	--	0.721
Male	49	47.6	136	49.6	--	

* standard deviation ** chi-squared test. MW: minimum wage.

**Table 2 healthcare-11-00948-t002:** PedsQL scores stratified into two groups: with pain and without pain.

	With Pain		Without Pain		Mean Diff.	Standardized Effect Size (95% CI) **	*p*-Value ***
	Mean	SD *	Mean	SD *			
Physical functioning	80.1	15.7	84	13.9	−3.9	0.25 (0.02–0.49)	0.017
Emotional functioning	56.9	23	66.7	20.8	−9.8	0.43 (0.20–0.67)	0.001
Social functioning	83.9	16.3	85	16.2	−1.1	-	0.485
School functioning	68.6	17.9	70.8	16.3	−2.2	-	0.368
Total score	73.6	14.9	78.5	12.7	−4.9	0.34 (0.10–0.58)	0.008

* standard deviation ** Cohen’s d *** chi-squared.

## Data Availability

All data generated during this study are available within this article.

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
