# Peer review of "The Impact of Low Back Pain on the Quality of Life of Children between 6 and 12 Years of Age"

_healthcare, 2023, doi:10.3390/healthcare11070948_

Round 1
Reviewer 1 Report
Review
First of all, I would like to congratulate the authors for the research conducted, and thank them for the opportunity to read their work. However, the present article has some major issues that should be addressed before its publication.
General comment:
Please, revise the writing style. Normally, impersonal style is more adequate for scientific dissemination than the personal style that can be identified in some parts of the present article.
Abstract:
The information of the abstract is adequate. The level of significant statistical differences considered does not seem to be necessary in this part. Perhaps it would be better to focus more in the results of the research.
Introduction:
The information of the introduction is scarce. I would recommend developing more the topic, including the etiology of back pain in the objective population, the possible factors that may affect this pathologies, a more detailed prevalence of back pain according to the region (cervical, dorsal, lumbar), and any other information that could help the reader to understand the need of the research. I would recommend the authors the following scientific literature that may help them in this task:
- González-Gálvez, N., Marcos-Pardo, P. J., Albaladejo-Saura, M., López-Vivancos, A., & Vaquero-Cristóbal, R. (2022). Effects of a Pilates programme in spinal curvatures and hamstring extensibility in adolescents with thoracic hyperkyphosis: a randomised controlled trial. Postgraduate medical journal, postgradmedj-2021-140901. Advance online publication. https://doi.org/10.1136/postgradmedj-2021-140901
- González-Gálvez N, Vaquero-Cristóbal R, López-Vivancos A, Albaladejo-Saura M, Marcos-Pardo PJ. Back Pain Related with Age, Anthropometric Variables, Sagittal Spinal Curvatures, Hamstring Extensibility, Physical Activity and Health Related Quality of Life in Male and Female High School Students. International Journal of Environmental Research and Public Health. 2020; 17(19):7293. https://doi.org/10.3390/ijerph17197293
- González-Gálvez N, Carrasco-Poyatos M, Vaquero-Cristóbal R, Marcos-Pardo PJ. Gender Mediation in Adolescents’ Back Pain and Physical Fitness: A Cross-Sectional Study. Healthcare. 2022; 10(4):696. https://doi.org/10.3390/healthcare10040696
Materials and Methods
The description of the sample provides all the information needed, including a flow diagram.
Respect to the instrument used, the authors state that it has been adapted and validated for their population. Could you, please, provide indicators of the validity and reliability of the Brazilian version of the questionnaire?
I would recommend the author to include a section in the materials and methods to describe the anthropometric measurements that they state in the statistical analysis.
If any other instrument has been used, as the Nordic Musculoskeletal Questionnaire, it should be described properly before the statistical analysis.
I would recommend the authors the inclusion of the odds ratio of suffering back pain depending on the variables analyzed.
I would recommend the authors to include the mean differences in the cases that the variables are quantitative, not only the statistical signification.
Discussion
As long as it has been stated in all the article, the author aimed to relate the back pain with the perceived quality of life. However, the first part of the introduction is focused on the activities of daily life limitations. Please, discuss the results of your study, as it is inappropriate to discuss variables that have not been measured, and establish a more argued and referenced relationship when talking about daily life limitations.
I would also recommend the inclusion of possible future lines of research.
Author Response
1) First of all, I would like to congratulate the authors for the research conducted, and thank them for the opportunity to read their work. However, the present article has some major issues that should be addressed before its publication.
We would like to thank the reviewer for their careful reading of the manuscript, detailed comments and suggestions.
2) General comment: Please, revise the writing style. Normally, impersonal style is more adequate for scientific dissemination than the personal style that can be identified in some parts of the present article.
We have followed the reviewer advice and changed all instances of first person writing to an impersonal style.
3) Abstract: The information of the abstract is adequate. The level of significant statistical differences considered does not seem to be necessary in this part. Perhaps it would be better to focus more in the results of the research.
We agree with the reviewer and have removed all p-values from the Abstract.
4) Introduction: The information of the introduction is scarce. I would recommend developing more the topic, including the etiology of back pain in the objective population, the possible factors that may affect these pathologies, a more detailed prevalence of back pain according to the region (cervical, dorsal, lumbar), and any other information that could help the reader to understand the need of the research. I would recommend the authors the following scientific literature that may help them in this task: […]
We have rewritten the introduction completely. As suggested, we have included information on low back pain (LBP) etiology in children and adolescents (page 2, lines 47 to 49), its possible risk factors in this population (page 2, lines 50 to 63) and life-course trajectory (page 2, lines 64 to 68). We have also highlighted the fact that consequences of LBP among children are still quite unknown (page 2, lines 71-80), especially when compared to the consequences of LBP in adults; we believe that in doing this we help the readers in understanding the importance of our research.
We would also like to thank the reviewer for their valuable literature recommendations, the articles have been very helpful and we have included them in the References.
5) Materials and Methods: Respect to the instrument used, the authors state that it has been adapted and validated for their population. Could you, please, provide indicators of the validity and reliability of the Brazilian version of the questionnaire?
We have included the Cronbach’s alpha value for the whole scale and for the individual domains. We have also explained that, even though the instrument has been in use for years in Brazil, knowledge about other psychometric properties of the Brazilian PedsQL 4.0 version is still lacking (page 3, lines 115 to 119).
6) Materials and Methods: I would recommend the author to include a section in the materials and methods to describe the anthropometric measurements that they state in the statistical analysis.
We have included a whole subsection dedicated to the anthropometric and kinesiologic measurements (page 4, lines 133 to 137). Nevertheless, since a detailed description of how each of the variables were measured have already been published [1], we have cited this publication in order to direct readers to it. We have done this to avoid any risk of duplicating previously published information and, thus, incurring into self-plagiarism.
[1] Santos EDS, Bernardes JM, Noll M, Gómez-Salgado J, Ruiz-Frutos C, Dias A. Prevalence of Low Back Pain and Associated Risks in School-Age Children. Pain Management Nursing. 2021;22(4):459-64.
7) Materials and Methods: If any other instrument has been used, as the Nordic Musculoskeletal Questionnaire, it should be described properly before the statistical analysis.
No other instrument was used in the research.
Regarding the Nordic Musculoskeletal Questionnaire (NMQ), it is important to note that only an adapted body map from the NMQ was used on this study in order to pinpoint the exact pain location, but we have not used the NMQ to obtain data. We understand that this was not explicit and may have caused confusion, thus we have included a statement to make it clearer (page 4, line 151 to 153).
8) Materials and Methods: I would recommend the authors the inclusion of the odds ratio of suffering back pain depending on the variables analyzed.
This manuscript’s aim was to investigate the relation of LBP and health-related quality of life (HRQoL), not to investigate LBP associated factors. However, a previous article with this objective has already been published [1]. Therefore, we have not included this information in this manuscript, since it would be considered as duplicate publication.
[1] Santos EDS, Bernardes JM, Noll M, Gómez-Salgado J, Ruiz-Frutos C, Dias A. Prevalence of Low Back Pain and Associated Risks in School-Age Children. Pain Management Nursing. 2021;22(4):459-64.
9) Materials and Methods: I would recommend the authors to include the mean differences in the cases that the variables are quantitative, not only the statistical signification.
We agree with the reviewer and have added not only the mean differences (page 5, tables 1 and 2) but also estimated the Cohen’s d effect size, a standardized difference of means, whereas the outcome (HRQoL) was significantly different when groups (with and without low back pain) were compared (page 5, table 2).
10) Discussion: As long as it has been stated in all the article, the author aimed to relate the back pain with the perceived quality of life. However, the first part of the introduction is focused on the activities of daily life limitations. Please, discuss the results of your study, as it is inappropriate to discuss variables that have not been measured, and establish a more argued and referenced relationship when talking about daily life limitations.
We would like to thank the reviewer for this valuable insight. We have mostly rewritten the discussion with this in mind.
As their suggestion, we have dedicated the discussion’s first part to the relation between LBP and HRQoL in children and to the magnitude of this relationship (page 6, lines 199 to 214). And even though we still discussed the possible relationship between activities of daily life (ADLs) limitations and the lower score in the HRQoL domain related to physical functioning in the group with low back pain (page 6, lines 215 to 229), we believe that we were able to establish a better reasoning and referenced relationship between our findings and ADLs. We consider that it was important to keep this topic in the discussion, since there is evidence that functional status’ impairments in children with low back pain may be associated to decreases in quality of life [1]. However, we agree with the reviewer that this had to be written in a more careful and referenced way, we hope that we had achieved it now.
[1] Balagué F, Ferrer M, Rajmil L, Pont Acuña A, Pellisé F, Cedraschi C. Assessing the association between low back pain, quality of life, and life events as reported by schoolchildren in a population-based study. European Journal of Pediatrics. 2012;171(3):507-14.
11) Discussion: I would also recommend the inclusion of possible future lines of research.
We agree with the reviewer and have included it in the manuscript (page 7, lines 263 to 271).
Reviewer 2 Report
Figure 1: In the "Children between" box, there should be n=990
Figure 1: In the "Interviewed children" box, there should be n=378
Figure 1: The last arrow should be between the "Interviewed children" and "Children included" boxes
Line 245: complete the citation 2022;19(14)…….
Line 252: complete the citation 2016;17(1)…….
Line 256: complete the citation 2021;13(21)…..
Line 268: complete the citation 2020;10(10)…..
Line 274: complete the citation 2013;……..
Line 300: complete the citation 2010;….
Line 302: complete the citation 2005;115(2)…..
It would be possible to know the incidence of low back pain due to its triggers or causes (trauma, abdominal, gynecological, urological problems,...) to assess how it influences the quality of life results.
Author Response
1) Figure 1: In the "Children between" box, there should be n=990
We apologize but we have not understood this comment, because there were not 990 children in the private school nor in the public schools. If the commentary is related to how the n information was presented, would like to express our thanks to the reviewer, because thanks to his careful reading we perceived that we had not standardized the way we presented this information. Therefore, we reviewed how it was done in recently published Healthcare’s articles and followed the pattern we found, which is an equal sign with a space after the letter “n” and another space before the number inside the parentheses (as can be seen in the figure - next page).
2) Figure 1: In the "Interviewed children" box, there should be n=378
We apologize but, unfortunately, we have not understood this comment as well, because there was already the “n = 378” information in the box. Again, if the commentary is related to how the n information was presented, would like to express our thanks to the reviewer, because thanks to his careful reading we perceived that we had not standardized the way we presented this information. Therefore, we reviewed how it was done in recently published Healthcare’s articles and followed the pattern we found, which is an equal sign with a space after the letter “n” and another space before the number inside the parentheses (as can be seen in the figure - next page).
3) Figure 1: The last arrow should be between the "Interviewed children" and "Children included"
We agree with the reviewer. However, once more, we have not understood the comment, since the last arrow is already between these boxes (as can be seen in the figure below).
4) Line 245: complete the citation 2022;19(14)
We have done it, thank you for pointing that out.
5) Line 252: complete the citation 2016;17(1)
We have done it, thank you for pointing that out.
6) Line 256: complete the citation 2021;13(21)
We have done it, thank you for pointing that out.
7) Line 268: complete the citation 2020;10(10)
We have done it, thank you for pointing that out.
8) Line 274: complete the citation 2013;
We have done it, thank you for pointing that out.
9) Line 300: complete the citation 2010;
We have done it, thank you for pointing that out.
10) Line 302: complete the citation 2005;115(2)
We have done it, thank you for pointing that out.
11) It would be possible to know the incidence of low back pain due to its triggers or causes (trauma, abdominal, gynecological, urological problems,...) to assess how it influences the quality of life results.
We have rewritten the introduction completely. As suggested, we have included information on low back pain (LBP) etiology in children and adolescents (page 2, lines 47 to 49), its possible risk factors in this population (page 2, lines 50 to 63) and life-course trajectory (page 2, lines 64 to 68). We have also highlighted the fact that consequences of LBP among children are still quite unknown (page 2, lines 71-80), especially when compared to the consequences of LBP in adults; we believe that in doing this we help the readers in understanding the importance of our research.
Reviewer 3 Report
This paper is more like a survey of characteristics of children with or without back pain. The most surprising finding of this study is that LBP is rising at an age range one would not expect.
All other findings seem correct, but they are more survey-like, rather than of explanatory nature, as the design of this study is cross-sectional. Therefore, no causal relationships can be inferred. Therefore, how can we be sure of what the authors imply about causal relationships, in lines 52-53, specifically: "Although the literature notes that psychosocial and emotional factors play important roles in the occurrence of low back pain", and further down in lines 164-168 also? I think that some obvious associations are highlighted, and other than that no revealing findings are presented.
Author Response
All other findings seem correct, but they are more survey-like, rather than of explanatory nature, as the design of this study is cross-sectional. Therefore, no causal relationships can be inferred. Therefore, how can we be sure of what the authors imply about causal relationships, in lines 52-53, specifically: “Although the literature notes that psychosocial and emotional factors play important roles in the occurrence of low back pain”, and further down in lines 164-168 also?
We agree with the reviewer that our study is not able to infer causal relations, due to its design. However, the information presented in lines 52-53, regarding low back pain risk factors, was not based on our results. This piece of information is in the introduction section, and it is a citation of a systematic review results. We even began the phrase with the following statement: “Although the literature notes […]”. Thus, it should be clear that we are not implying any causal relation based on our results.
The same thing is true for the phrase that begins in line 164. It is a citation of the results from another systematic review, that we used to back up an explanatory hypothesis to one of our findings. We even took the care to start the phrase stating that it was a “possible explanation”.
I think that some obvious associations are highlighted, and other than that no revealing findings are presented.
We must beg to differ on this. Our study presents a novelty of great importance, the investigation of the relation between low back pain (LBP) and health-relate quality of life (HRQoL) among children between 6 and 12 years of age. This is a younger age range than those investigated in previous studies regarding this topic, most of them studied children older than 12 years of age.
Regarding our results, they show a significant difference in HRQoL and in two HRQoL domains (physical and emotional functioning) of moderate magnitude between these young children with and without LBP, while previous studies had found only a low effect of LBP in HRQoL [1,2]. This is quite important, since the specific impact of LBP among children has not been fully studied yet and even less so in such young ones. Our results are also quite valuable because they may be pointing out that (1) the relation between LBP and HRQoL may be of a higher magnitude among younger children; and/or that (2) this relation’s magnitude has increased in the last decade, since the previous studies that found only a low effect of LBP in HRQoL are from 2009 and 2012.
[1] Balagué F, Ferrer M, Rajmil L, Pont Acuña A, Pellisé F, Cedraschi C. Assessing the association between low back pain, quality of life, and life events as reported by schoolchildren in a population-based study. European Journal of Pediatrics. 2012;171(3):507-14.
[2] Pellisé F, Balagué F, Rajmil L, Cedraschi C, Aguirre M, Fontecha CG, et al. Prevalence of Low Back Pain and Its Effect on Health-Related Quality of Life in Adolescents. Archives of Pediatrics & Adolescent Medicine. 2009;163(1):65-71.
Reviewer 4 Report
1. Did you correlate school performance with back pain
2. Did you review outside recreational activities and back pain
3. Public and Private schools- was that dependent on economic status for private school
4. Justify use of PedsQol 4.0 as indicator
5. What was the outcome of the back pain?
6. Did you explore cyber bullying as cause of back pain?
7. Were there any siblings with back pain?
8. Any history of depression in subjects. parents or siblings?
Author Response
- Did you correlate school performance with back pain
This study aimed to investigate the impact of low back pain (LBP) on the perceived health-related quality of life (HRQoL), not to investigate LBP associated factors, that is why we have not investigated this factor. However, a previous article with this objective has already been published [1]. We have not included this information in this manuscript, because it would be considered a duplicate publication.
[1] Santos EDS, Bernardes JM, Noll M, Gómez-Salgado J, Ruiz-Frutos C, Dias A. Prevalence of Low Back Pain and Associated Risks in School-Age Children. Pain Management Nursing. 2021;22(4):459-64.
2) Did you review outside recreational activities and back pain
We have not studied recreational activities. However, we would like to remember that this was not a study that aimed to find LBP and/or HRQoL associated factors, our objective was to investigate the relation of LBP and HRQoL, through groups (children with LBP and children without LBP) comparison.
3) Public and Private schools- was that dependent on economic status for private school.
Yes, it was. However, it is important to note that our results show no significant difference regarding the occurrence of LBP due to income.
4) Justify use of PedsQol 4.0 as indicator
The PedsQL has been in use since 2001 for numerous pediatric health conditions, including LBP. It has demonstrated adequate reliability, validity, sensitivity and responsiveness for child self-report for ages 5–18 years. It has also been shown that it is related to other key constructs in pediatric healthcare such as access to needed care, healthcare barriers and quality of primary care [1]. Finally, it had already been translated and validated to Brazilian Portuguese [2].
[1] Varni JW, Burwinkle TM, Seid M. The PedsQL as a pediatric patient-reported outcome: reliability and validity of the PedsQL Measurement Model in 25,000 children. Expert Ver Pharmacoecon Outcomes Res. 2005 Dec;5(6):705-19.
[2] Klatchoian DA, Len CA, Terreri MTRA, Silva M, Itamoto C, Ciconelli RM, et al. Qualidade de vida de crianças e adolescentes de São Paulo: confiabilidade e validade da versão brasileira do questionário genérico Pediatric Quality of Life InventoryTM versão 4.0. J pediatr (Rio J). 2008;84(4):308-15.
5) What was the outcome of the back pain?
Since this was a cross-sectional study, we were not able (and it was not our objective) to evaluate the clinical course and outcome of participants with LBP. However, according to the literature, it is expected that most cases of LBP in children will have a benign course.
6) Did you explore cyber bullying as cause of back pain?
This study aimed to investigate the impact of LBP on the perceived HRQoL, not to investigate LBP associated factors, that is why we have not investigated this factor. However, a previous article with this objective has already been published [1]. We have not included this information in this manuscript, because it would be considered a duplicate publication.
[1] Santos EDS, Bernardes JM, Noll M, Gómez-Salgado J, Ruiz-Frutos C, Dias A. Prevalence of Low Back Pain and Associated Risks in School-Age Children. Pain Management Nursing. 2021;22(4):459-64.
7) Were there any siblings with back pain?
We have not studied history of low back pain in siblings. However, we would like to remember that this was not a study that aimed to find LBP and/or HRQoL associated factors, our objective was to investigate the relation of LBP and HRQoL, through groups (children with LBP and children without LBP) comparison.
8) Any history of depression in subjects. Parents or siblings?
We have not studied subjects nor familiar’s history of depression. However, we would like to remember that this was not a study that aimed to find LBP and/or HRQoL associated factors, our objective was to investigate the relation of LBP and HRQoL, through groups (children with LBP and children without LBP) comparison.
Reviewer 5 Report
Dear authors,
I read the manuscript with interest.
Unfortunately I have detected a serious problem that will be difficult to overcome. The data collected dates back to 2017, more than 5 years have already passed so it may not reflect the current population.
Other critical points I will list below:
-1) In the introduction, many of the statements refer to adults. I recommend limiting yourself to the considerations on children by expanding the references.
2) the substantial reference to lifestyle is missing. Tablets, computers, video games, a sedentary lifestyle favor low back pain.
3) In the discussion there are statements that are not proved by the results.
4) I recommend reviewing the results but above all the discussion.
5) ietms not present in the results (HRQOL) appear in the discussion.
6) Criticality:
Study time 2017.
No concomitant pathologies have been investigated (they could favor the onset of low back pain).
Pathologies such as dysmorphisms and paramorphisms
Systemic pathologies
Sports favoring low back pain have not been investigated (they may alter the results).
No risk factors were investigated.
Author Response
1) I read the manuscript with interest.
We would like to thank the reviewer for the valuable suggestions.
2) Unfortunately I have detected a serious problem that will be difficult to overcome. The data collected dates back to 2017, more than 5 years have already passed so it may not reflect the current population.
We do understand and in normal circumstances we would wholeheartedly agree with the reviewer’s comment. However, it is important to note that there is no evidence of impactful changes in school routine and/or children behavior when comparing the data collection period and the current moment. In addition, in Brazil we had more than 30 months without in-person classes due to the Covid-19 pandemic, so despite the data being from 2017, we believe that they do not reflect a different reality from the current one.
We also have no reason to believe that the relation that we have found between LBP and HRQoL has suffered any changes from 2017 to nowadays.
3) In the introduction, many of the statements refer to adults. I recommend limiting yourself to the considerations on children by expanding the references.
We have rewritten the introduction completely. We believe that now it is much more focused on the study’s population. However, we kept some information regarding LBP in adults in order to demonstrate how much more we know about LBP and its consequences in adults than in children.
4) the substantial reference to lifestyle is missing. Tablets, computers, video games, a sedentary lifestyle favor low back pain.
This study aimed to investigate the impact of LBP on the perceived HRQoL, not to investigate LBP associated factors, that’s why we have not investigated these factors. However, a previous article with this objective has already been published [1]. We have not included this information in this manuscript, because it would be considered a duplicate publication.
[1] Santos EDS, Bernardes JM, Noll M, Gómez-Salgado J, Ruiz-Frutos C, Dias A. Prevalence of Low Back Pain and Associated Risks in School-Age Children. Pain Management Nursing. 2021;22(4):459-64.
5) In the discussion there are statements that are not proved by the results.
We would like to thank the reviewer for this valuable insight. We have mostly rewritten the discussion with this in mind.
We have dedicated the discussion’s first part to the relation between LBP and HRQoL in children and to the magnitude of this relationship (page 6, lines 199 to 214). And even though we still discussed the possible relationship between activities of daily life (ADLs) limitations and the lower score in the HRQoL domain related to physical functioning in the group with low back pain (page 6, lines 215 to 229), we believe that we were able to establish a better reasoning and referenced relationship between our findings and ADLs. We consider that it was important to keep this topic in the discussion, since there is evidence that functional status’ impairments in children with low back pain may be associated to decreases in quality of life [1]. However, we agree with the reviewer that this had to be written in a more careful and referenced way, we hope that we had achieved it now.
[1] Balagué F, Ferrer M, Rajmil L, Pont Acuña A, Pellisé F, Cedraschi C. Assessing the association between low back pain, quality of life, and life events as reported by schoolchildren in a population-based study. European Journal of Pediatrics. 2012;171(3):507-14.
6) I recommend reviewing the results but above all the discussion.
As we have answered in the last comment, we have mostly rewritten the discussion. We have also revised our results and we have added not only the mean differences (page 5, tables 1 and 2) but also estimated the Cohen’s d effect size, a standardized difference of means, whereas the outcome (HRQoL) was significantly different when groups (with and without low back pain) were compared (page 5, table 2).
7) items not present in the results (HRQOL) appear in the discussion.
As we have answered in the last comment, we have mostly rewritten the discussion with this in mind, we would like to express our thanks for pointing it out to us.
8) No concomitant pathologies have been investigated (they could favor the onset of low back pain). Pathologies such as dysmorphisms and paramorphisms. Systemic pathologies.
This study aimed to investigate the impact of LBP on the perceived HRQoL, not to investigate LBP associated factors, that’s why we have not investigated these factors. However, a previous article with this objective has already been published [1]. We have not included this information in this manuscript, because it would be considered a duplicate publication.
[1] Santos EDS, Bernardes JM, Noll M, Gómez-Salgado J, Ruiz-Frutos C, Dias A. Prevalence of Low Back Pain and Associated Risks in School-Age Children. Pain Management Nursing. 2021;22(4):459-64.
9) Sports favoring low back pain have not been investigated (they may alter the results).
This study aimed to investigate the impact of LBP on the perceived HRQoL, not to investigate LBP associated factors, that’s why we have not investigated this factor. However, a previous article with this objective has already been published [1]. We have not included this information in this manuscript, because it would be considered a duplicate publication.
[1] Santos EDS, Bernardes JM, Noll M, Gómez-Salgado J, Ruiz-Frutos C, Dias A. Prevalence of Low Back Pain and Associated Risks in School-Age Children. Pain Management Nursing. 2021;22(4):459-64.
10) No risk factors were investigated.
This study aimed to investigate the impact of LBP on the perceived HRQoL, not to investigate LBP associated factors. However, a previous article with this objective has already been published [1]. We have not included this information in this manuscript, because it would be considered a duplicate publication.
[1] Santos EDS, Bernardes JM, Noll M, Gómez-Salgado J, Ruiz-Frutos C, Dias A. Prevalence of Low Back Pain and Associated Risks in School-Age Children. Pain Management Nursing. 2021;22(4):459-64.
Round 2
Reviewer 1 Report
I would like to thank the authors the efforts to improve the manuscript.
Author Response
Dear Reviewer,
thank you for your comments and decision.
Reviewer 3 Report
Dear authors,
Regarding my first comment, yes, it was apparent in both parts of the manuscript you were referring to previous studies which were not yours. What I meant was that the design of those previous studies was most likely prospective and not cross-sectional like yours. Therefore, no direct comparisons can be made between prospective and cross-sectional studies.
However, I must note now that the manuscript is improved, with valuable additions in the introduction section, clarifying further the significance of LBP research in children. Also, the statistics of the study are improved (including standardized effect sizes), and there is now relevant discussion regarding the strength of the findings of the current study.
Author Response

(The authors gave the same response as above.)

Reviewer 5 Report
Thanks for making the changes. However, the issue of recruitment for me is insurmountable. Giving information dating back to 2017 does not seem correct to me. Therefore you advise to reject the manuscript.Author Response
Dear Reviewer,
thank you for your comments.